# Unveiling Network Performance in the Wild: An Ad-Driven Analysis of Mobile Download Speeds

## ABSTRACT

Accurate measurement of mobile network performance is crucial for optimizing user experience and ensuring regulatory compliance. Traditional methods like crowdsourcing approaches, though effective, depend heavily on user participation and extensive infrastructure. In this paper, we introduce *adNPM*, a novel technique for measuring download speed by embedded measurement code in ads displayed across web browsers and mobile apps, without requiring user participation. Through controlled lab tests and real-world deployments in 15 countries, we show that *adNPM* produces results comparable to well-established tools such as Speedtest by Ookla and Opensignal while consuming significantly less data.

Our solution leverages ad campaigns to collect extensive data from diverse demographics and geographic regions, providing deep insights into the performance of major Internet Service Providers (ISPs). Furthermore, *adNPM* can segment download speed analyses by demographic factors and operating systems, making it a versatile and scalable tool for network performance assessment.

## KEYWORDS

d/l speed, network measurements, ad, adTag, bandwidth

## 1 INTRODUCTION

In the last decade, the architecture of Internet services has evolved towards a centralized infrastructure (video streaming, social networks, etc.) services sitting in the core network infrastructure (data centers, CDNs, etc.) which are massively consumed by end users. One of the most crucial performance parameters for end-users is bandwidth and, more specifically, the download bandwidth provided by their connection. In essence, this parameter determines the range of services a user can access, from basic web browsing and standard video streaming (SD) or audio streaming, requiring 1-5 Mbps, to more bandwidth-intensive activities such as playing first-person shooter (FPS) high-resolution video games, which typically demand 25-50 Mbps.

Several methodologies for measuring the speed of end-users' network connections have been developed. But most rely on active user participation, on complex infrastructure or dedicated applications, which may introduce biases and limit global reach. Currently, the most readily available tools for such assessments are commonly known as *Speed Tests*. These tools, whether in the form of web-pages or mobile applications, enable users to voluntarily measure their connection speeds by transmitting a significant amount of data between a server and the user's device in both directions, measuring upload and download bandwidth. While effective, their coverage relies on user participation.

In this paper, we present *adNPM*, a novel technique that opportunistically measures the download speed from code embedded in ads. We set up an advertising campaign that includes our measurement code within the ads. Each time our ad is rendered on a device (in a browser or mobile app), our measurement code is executed inside the ad to measure the download speed of the link. Unlike existing solutions, *adNPM* does not demand volunteers. Moreover, it facilitates deploying targeted measurement campaigns since advertising platforms allow for defining targeting parameters across different dimensions: (i) location; (ii) demographics; (iii) type of device (mobile vs. fixed); (iv) operating system & browser; etc. Furthermore, *adNPM* proves to be a very cost-effective method, as it leverages the existing advertising infrastructure to achieve large-scale measurements with minimal additional expenditure for research.

We evaluate the performance of our technique in a lab-controlled environment, consider different cross-traffic scenarios, and compare it with existing commercial speed measurement solutions. We also compare the reported speed at the country level between our solution and those reported by Speedtest by Ookla, one of the most popular commercial solutions, and Opensignal, an independent source for quantifying mobile network quality, both based on billions of measurements in the wild of actual mobile network link speeds. Our measurements in the wild show that *adNPM* achieves accuracy levels similar to these established tools while using significantly less data. This thorough evaluation concludes that our solution provides reliable and accurate measurements, making it a valuable tool for assessing mobile network performance.

Finally, we run real advertising campaigns to assess the speed offered by the main infrastructure-based Internet Service Providers (ISPs) in 15 countries. Moreover, the targeting properties offered by advertising platforms allow us to report speed differences observed across different demographic groups based on ages and gender as well as differences offered by different operating systems (Android vs. iOS).

## 2  BACKGROUND

### 2.1  Mobile network download bandwidth

Approaches for measuring mobile network performance include *field testing*, *testbed-based research platforms*, *automated mobile bandwidth measurement*, and *crowdsourcing-based mobile bandwidth measurement*.

Field testing offers insights into mobile network performance in real-world conditions [38, 44, 92, 93], but its high costs and scalability limitations make it impractical for large-scale or geographically extensive assessments [16, 65].

Testbed platforms provide controlled environments for precise network measurements [3, 61, 102], but their lack of real-world conditions and the high costs of setup and maintenance limit their ability to capture the dynamic nature of mobile network performance at scale.

Automated measurement platforms like Opensignal [83] collect data passively from millions of devices, but their reliance on extensive infrastructure and limited demographic targeting reduce scalability and may not capture performance variations across different user groups. Opensignal reports average download speeds by aggregating data from major ISP, so it requires accurate market share information to ensure representative results.

Crowdsourced bandwidth measurements rely on voluntary user participation [2, 39, 59, 64, 89], which may limit accessibility and lead to unrepresentative results due to specific test conditions and varying technical expertise. Moreover, the dependence on user initiation may exclude groups with less technical expertise, which reduces the overall completeness of the data. Furthermore, the dependence on user initiation may exclude less tech-savvy groups, reducing the overall data comprehensiveness. Tools like Speedtest by Ookla [17, 24] and nPerf [79] provides measurements supported by a global network of servers to route user traffic to the nearest servers, while M-Lab (Measurement Lab) [70] and SpeedSmart [95] use dedicated testing infrastructure, which may not fully reflect common Internet traffic. Fast.com [48] uses Netflix's own servers.

### 2.2  Ad-based measurements

Ads are frequently used as a venue for embedding measurement scripts in online advertising [28], enabling experiments to be conducted each time the ad is rendered. Advertisers and third-party providers add scripts to monitor KPIs, detect fraudulent activities, and interact with backend servers [78]. The research community has also leveraged ads to analyze transparency in online advertising [47, 72], measure ad blockers' impact [72], assess DNS performance [27], optimize content delivery [63, 101], and evaluate energy consumption [56, 88] or device vulnerability to web fingerprinting [18].

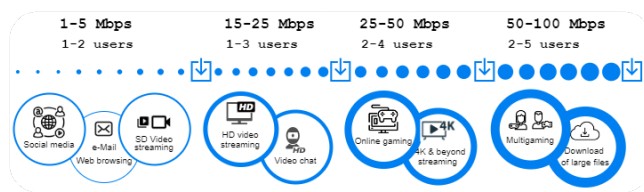

**Figure 1: Suggested speeds for popular online activities.**

The online advertising ecosystem, driven by automated real-time bidding, allows for targeted campaigns based on location, demographics, and device type, which also enables precise, targeted ad-based measurement experiments. Various ad formats, including video, graphical, and search ads, offer flexibility for such measurements.

In Section 3.1, we describe our ad-based approach to measuring mobile network performance through download speed.

### 2.3  Speed contextualization

The download speed of a mobile connection is crucial in determining which online services a device can access, as different services have varying bandwidth requirements. Figure 1 summarizes the recommended download speeds for accessing popular services, ranging from 1 Mbps to 100 Mbps [4, 33, 57, 76]. This provides two key points of reference. Firstly, by measuring the download speed of a connection, we can assess which services are accessible through that connection. For example, speeds of 15-25 Mbps enable HD streaming and video chat for 1-3 users. Secondly, due to the wide range of speeds (1 to 100 Mbps), it is essential to have an accurate measurement technique. For instance, a 25% measurement error in a 17 Mbps connection, reporting a speed of 12.75 Mbps, would wrongly suggest that the connection is unsuitable for HD streaming. Instead, a 25% error in a 200 Mbps connection, leading to a measurement of 150 Mbps, has no practical impact on the usability of services.

The Federal Communications Commission (FCC) of the United States recommends a download internet rate of 12-25 Mbps for families with multiple internet users [33]. Therefore, a download speed of at least 100 Mbps is widely considered fast enough to handle nearly any online activity in an average household with typical bandwidth usage. For instance, streaming 4K high-definition videos on multiple devices, watching Netflix or YouTube, attending Zoom meetings, downloading large files for work, or playing HD games.

## 3  METHODOLOGY

### 3.1  *adNPM*: description methodology

*adNPM* assess network performance throughput by measuring download bandwidth on mobile network connections

using ads displayed in web browsers and mobile apps, specifically on Android and iOS. This approach uses advertising tags (*adTag*) to overcome the limitations of conventional measurement techniques described in Section 2, as the strength of online advertising allows reaching different demographics in different locations with similar or diverse interests, across various platforms and devices, without relying on user engagement or the need for extensive infrastructure setup. To measure the download speed from a device, we develop our *adTag*, a JavaScript code embedded in any ad format capable of including a script. Our *adTag* is executed during the ad rendering process on a device. Note that *adNPM* can also measure fixed network performance -wired or WiFi- on both desktop and mobile devices.

The designed ad is a video display creative with a size (S) of 5,5 MB (5742549 Bytes), configured with no auto-play function. We define this size to measure download speeds of up to 110 Mbps, which covers the speed range discussed in Section 2. Larger video sizes would allow us to measure faster connection speeds. However, they would also consume more bandwidth and impose a higher overhead on the device. Note that, in the worst-case scenario, *adNPM* imposes a maximum download size of 65.72 MB of data for a reliable speed measurement of 110 Mbps. Significantly lower than Speedtest by Ookla/SpeedSmart/Fast.com at similar speed: 223/177/111 MB.

## 3.2 *adNPM* workflow

Once our ad is rendered in a user's browser (or mobile app), our *adTag* is executed and retrieves the link to the video object, downloading it in the background via the browser's *fetch* API. To obtain a statistically meaningful download speed measurement, *adNPM* performs 12 fetches, divided into two groups of 6, complying with the policies and restrictions imposed by modern browsers (e.g., Google Chrome, Safari, Firefox, Edge, Opera, Android, iOS, and IE Mobile) on the number of simultaneous connections to a single domain, which is typically 6 [40]. The second group of fetches is triggered as soon as a fetch from the first group completes; at that point, a fetch from the second group is automatically initiated, ensuring incremental and proportional data downloading. This approach allows meaningful network speed measurement even in low-to-medium-bandwidth connections. For instance, at 5 Mbps (10 Mbps), *adNPM* downloads 6.35 MB (10.5 MB) of data. *adNPM* sets a maximum time limit of 7.5 seconds for all fetch attempts to complete. Download speed measurement is computed based on all the triggered fetches.

Each fetch $f_i$ leads to a download speed sample ($\mathcal{DS}_i$), as:

$$\mathcal{DS}_i = \frac{S_i}{T_i} \quad (1)$$

**Table 1: *adNPM* and crowdsourcing tools performance across 2 to 110 Mbps range, with 5 Mbps intervals.**

| Measurements | *adNPM* | Ookla | M-Lab | SpeedSmart | nPerf | Fast.com |
|---|---|---|---|---|---|---|
| MRE | 3.49% | 5.50% | 8.57% | 1.60% | 3.55% | 11.40% |
| $\varepsilon$ | 1.65 | 3.25 | 5.10 | 1.03 | 1.47 | 7.04 |
| Duration (s) | 7.5 | 20 | 10 | 15 | 23 | 8 |
| Avg. data Xfer (MB) | 46.75 | 118.22 | | 93.18 | | 58.55 |

where $S_i$ is the download chunk size and $T_i$ is the time taken.

The $\mathcal{DS}$ of the connection is calculated as the average of the fetches $f_i$ that either fully download the video object or retrieve the largest portion of it. Fetches that do not provide stable or representative measurements of typical network conditions, such as those that complete unusually fast or slow, are discarded. By limiting the calculation to these selected fetches, we achieve more reliable and consistent results, avoiding issues related to incomplete or unstable fetches.

## 3.3 Methodology validation

We evaluate *adNPM* accuracy through extensive lab experiments, comparing its performance with state-of-the-art speed measurement tools introduced in Section 2.

*3.3.1 Controlled lab experiments.* In a lab set-up, we connect a mobile device to the Internet via a computer acting as a router, which limits the download speed to predefined values for ground-truth comparison. We deploy different speed measurement tools, including *adNPM*, and compare the reported download speeds to these ground-truth values to assess error rates. In addition, to simulate real-world scenarios, we introduce low-level background traffic, as typical mobile apps (e.g., streaming, email, location services, or weather apps) generate some data while in the background [31, 67, 75, 109]. This allows us to assess the accuracy of different tools under realistic cross-traffic conditions.

*Lab tests.* We emulate 22 connection speeds, ranging from 2 to 110 Mbps, with 5 Mbps intervals[1]. For each connection speed and measurement tool, we conduct three measurement experiments (66 total), computing the Mean Relative Error (MRE) and Mean Absolute Error ($\varepsilon$) by comparing the average of measured speed with the ground truth. We also record the measurement duration and average data transferred. As shown in Table 1, *adNPM* boasts reliable accuracy, ensuring an MRE below 3.5% and $\varepsilon$ of 1.65, and performs faster than the other tools. SpeedSmart shows the lowest MRE (1.60%) and $\varepsilon$ (1.03), due to its global server network, which operates independently from the broader network infrastructure for instrumentation. While tools like M-Lab and

---

[1]Monitored connection speed starts at 2 Mbps; next is 5 Mbps; from there, increasing by 5 Mbps at a time until reaching 110 Mbps.

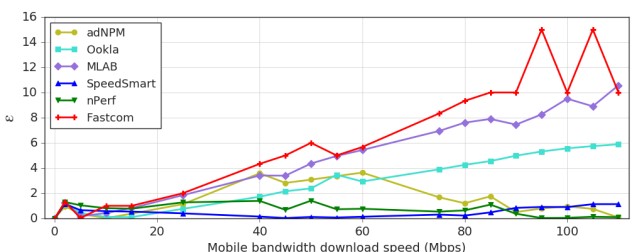

**Figure 2: Precision by absolute error ($\varepsilon$) of *adNPM* and speed tools under cross-traffic (see in color).**

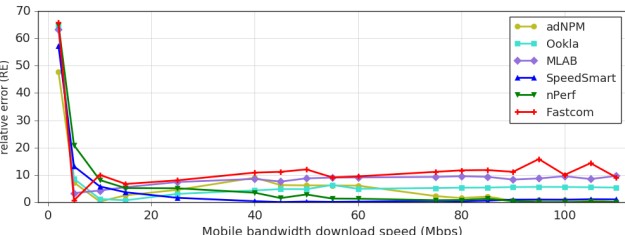

**Figure 3: Precision by relative error (RE) of *adNPM* and speed tools under cross-traffic (see in color).**

Fast.com have higher variability and slightly lower accuracy. nPerf achieves similar results to *adNPM* although takes longer ($\approx$ 25 seconds). Speedtest by Ookla performs well, but with a higher MRE (+2%) and double $\varepsilon$, requiring $\approx$ 20 seconds for measurement.

*adNPM* accomplishes these measurements using roughly half the data downloaded to the user's device. For bandwidth with a download speed of $\approx$ 25 Mbps ($\approx$ 50 Mbps), *adNPM* consumes 27.3 MB (52.5 MB) versus 48.3 MB (102 MB) for Speedtest by Ookla. Note that our technique imposes a maximum download limitation of 65.72 MB. This means that for download speeds exceeding 70 Mbps, the data transferred on the user's device is capped at 65.72 MB. Despite this limitation, *adNPM* reliably measures speeds up to 110 Mbps, while Speedtest by Ookla continues to proportionally flood the internet connection of the user's device. For a speed of $\approx$ 110 Mbps, Speedtest by Ookla generates a download of 223 MB.

*Lab tests with cross-traffic.* We replicate the previous experiments (with link speeds between 2 and 110 Mbps), inserting realistic cross-traffic patterns using iPerf3 [46] to simulate background traffic from common mobile applications such as streaming platforms, GPS location and navigation services, weather applications, email notifications, and application updates, among others. This setup mimics the typical behavior of mobile devices, stressing the network by sending packets to saturate the bandwidth while conducting performance tests to evaluate the download speed of the connection, based on prior studies of mobile traffic patterns [4, 23, 31, 36, 84, 85, 94, 109].

Figure 2 and Figure 3 show the absolute error ($\varepsilon$) and relative error (RE) for *adNPM* and other tools (Speedtest by Ookla, M-Lab, nPerf, SpeedSmart, Fast.com). *adNPM* maintains accuracy with absolute errors under 4 Mbps and a mean relative error (MRE) of 5.81%. While SpeedSmart and nPerf perform better at medium speeds, they are more susceptible to performance variations at low download speeds due to cross-traffic imposes more load on the bandwidth. They give MRE values of 4.91% and 6.56%, respectively. Ookla, M-Lab,

and Fast.com show larger measurement errors, with MRE values growing at faster speeds: MRE ($\varepsilon$) of 8.14% (6), 11.06% (10) and13.25% (+14 ) Mbps, respectively.

## 3.4 Limitations

Our methodology focuses on measuring network download speed on mobile devices, unlike other approaches that include upload speed. However, download speed is the most critical metric in today's internet usage, as most online activities—such as web browsing, streaming, and content downloads—rely heavily on fast downloads for a smooth user experience. Downlink traffic accounts for over 90% of total traffic in mobile networks, with video applications taking up 97% of traffic in that direction [90]. Therefore, measuring download speed is essential for assessing user experience and coverage across different areas by Internet Service Providers (ISPs) [34, 48, 77, 83].

## 4 DOWNLOAD SPEED IN THE WILD

### 4.1 Experiment set-up

We deploy *adNPM* in real advertising campaigns via the Sonata Platform, a Demand Side Platform (DSP) operated by TAPTAP Digital [42]. Sonata is a mid-sized DSP that serves tens of millions of daily ads in 15 countries across Europe, Africa, and the Americas. Our display video ad is hosted on TAPTAP's CDN (Content Delivery Network) to ensure proximity to the device and reduce potential artifacts from distant server downloads.

Sonata's targeting capabilities allows to focus on location, demographics (age, gender), device type (mobile vs. fixed), or Operating System. In the mobile realm, as the OS market is unevenly dominated by Android (71.5%) and iOS (28%) [98], we use Sonata DSP's targeting capabilities to set up ad campaigns for each OS individually, but for space constraints, they are presented grouped in Table 2. Moreover, to ensure adequate reported results to OSes market share, we weight our dataset according to that market share. For geographic targeting, we select 15 countries where Sonata operates, ensuring a diverse sample across three continents

and different development levels. Each country have at least 300 unique speed measurement samples, meeting Ookla's criteria for valid country-level reporting [25]. Additionally, we use Sonata's targeting capabilities to extract demographic data (age, gender) when available.

Finally, we collect the ISP associated with each sample from the `deviceCarrier` attribute reported by our DSP.

*4.1.1 Presence of outliers in Measurements in the wild.* adNPM may encounter limitations in environments with excessive cross-traffic, congested networks, or spurious failures. To mitigate this potential risk, as Ookla and other tools, we use the median as a baseline measure, as it is less affected by outliers, ensuring a more accurate representation of the central tendency of our measurements under typical network conditions. Furthermore, just as the other measurement bandwidth tools filter out extreme values to maintain the integrity of their data [74, 83], we discard samples where the aggregate download `duration` exceeds 99.5 seconds, ensuring a 10% margin over our 7.5 second threshold for each fetch. This filtering ensures that unusually long loading times, such as network anomalies or extremely low bandwidth connections, do not skew results.

*4.1.2 Data cleaning and curation.* To address the variability and imbalance in the market share of mobile OS across targeted countries, we adopt a rigorous approach to present our internet download speed results as more accurately reflecting the average user's experience. For instance, in Italy, 67.32% of samples are Android (32.27% are iOS), while in Argentina, Android represents 90.01% (9.8% iOS)[2]. Thus, we calculate the download speed metric for each operating system separately before aggregating the results, based on the market share of each considered country.

*4.1.3 Measurements in the wild.* While considered crowd-sourcing tools provide accurate measurements (see 3.3.1), we focus on Speedtest by Ookla due to its canonical status and widespread usage. Ookla shares a comprehensive speed rankings for over 170 regions. We also incorporate Opensignal, which specializes in mobile network quality and provides detailed operator-level data. By considering the market share of mobile networks, we can find out average download speeds per country. Moreover, as detailed in Section 5, we conduct demographic analyses by gender and age, although Ookla and Opensignal do not tackle this scope of analyses.

*4.1.4 Effectiveness of* adNPM *Measurement in the wild.*

---

[2]Average mobile OS market share reported by Statcounter Global Stats [97] for the period of Oct-Dec 2023, which aligns with the experiment period.

**Table 2: Ad campaigns used to populate *adNMP* dataset.**

| ID | Time Frame | Ad Source | OS | Delivery ad impressions | Launched measure. | Valid measure. |
|----|-----------|-----------|-----|------------------------|-------------------|----------------|
| 01 | Oct 18-22 '23 | browsers | Android; iOS | 369182 | 333728 | 202621 |
| 02 | Nov 17-22 '23 | browsers | Android; iOS | 1200205 | 1131022 | 748505 |
| 03 | Nov 24-29 '23 | apps | Android; iOS | 1200187 | 1031080 | 889547 |
| 04 | Dec 11-16 '23 | browsers | Android; iOS | 1200120 | 735223 | 428289 |
| 05 | Dec 18-23 '23 | apps | Android; iOS | 1200468 | 660545 | 557474 |

*Web browser measurements.* Several factors can disrupt download speed measurements using our methodology. Specifically: 1) Google's heavy ad intervention policies [54], impacting 13% of web browser samples; 2) errors during ad loading or execution, affecting 10%; and 3) issues such as ad blockers [106], disabled JavaScript [41], outdated browsers, limited browser API support, and network issues between the ad and our server. After excluding cases where the code is not executed, *adNPM* successfully completes measurements in 63% of browser cases.

*Mobile apps.* Mobile app measurements are not affected by Google's policies [54], and only 1% of samples encounter errors during ad retrieval. As a result, adNPM achieves a success rate of 86% in mobile apps.

## 4.2 Measurements and Datasets

We use Sonata Platform to run *adNPM*-enabled ad campaigns between October and December 2023. Table 2 outlines each campaign's time frame, measurement source (browsers vs. apps), number of ad impressions delivered, *adNPM* executions, and valid measurements.

While our campaigns target mobile devices, the Sonata Platform does not differentiate based on connection technology: cellular and WiFi connections. To address this, our adTag uses the Network Information API [104] to identify connection types, though it provides data for roughly 60% of the measurement samples. For the remaining samples, we apply an imputation technique based on IP prefixes identified as cellular, making the assumption that fixed and cellular IP prefixes are generally independent, which is supported by previous studies [37, 45, 103]. We collect the IP address and prefix of the connection identified as cellular by the Network Information API. We then seek those samples belonging to the identified cellular IP prefixes in the not-labeled measurement set. After data curation and applying our imputation technique, our dataset includes 418k measurement samples of cellular network links.

The total cost of campaigns is 410.26 €, resulting in a cost of $1.45 \times 10^{-4}$ € per valid measurement. This demonstrates that adNPM is a cost-effective, scalable method for measuring download speeds, without relying on volunteers (e.g.,

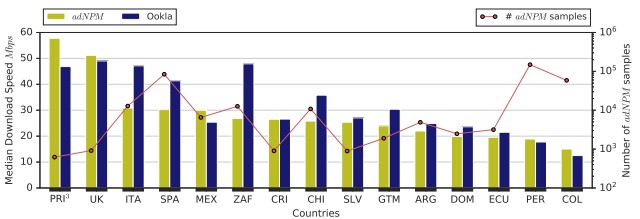

**Figure 4:** *left* y-axis: *adNPM* and Ookla median d/l speed by country; *right* y-axis: ad campaign samples.

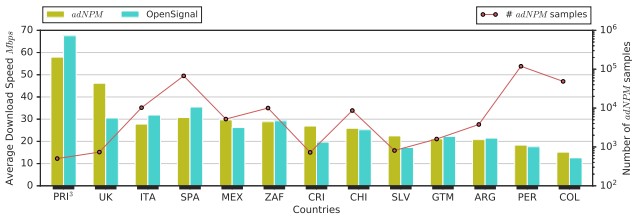

**Figure 5:** *left* y-axis: *adNPM* and Opensignal average d/l speed by country; *right* y-axis: ad campaign samples.

Speedtest by Ookla) or large-scale deployed infrastructure (e.g., Opensignal).

We plan to publicly release our dataset to encourage further research and validation.

## 5 RESULTS

In this section, we present the download speed results from *adNPM* across 15 countries. We analyze: 1) the download speeds offered by main mobile operators; 2) whether there exists any bias in speed based on demographic factors, which is critical for equitable Internet access [60]; and 3) the impact of Android vs. iOS on download speeds.

### 5.1 Country-level Download Speed

To understand the impact of the development factor on the mobile network speed of a country, we compute Spearman's rank correlation coefficient between download speeds from *adNPM* and the GDP per capita [14]. The strong reported correlation (0,95) suggests that countries with higher economic and technological development tend to have better download speeds.

We also compare *adNPM*'s results with publicly available data from Speedtest by Ookla and Opensignal.

Figure 4 shows the median download speeds measured by *adNPM* at the country-level: Puerto Rico[3], United Kingdom, Italy, Spain, Mexico, South Africa, Costa Rica, Chile, El Salvador, Guatemala, Argentina, Dominican Republic, Ecuador, Peru, and Colombia. In Figure 4, we present the average monthly median speed value from October to December 2023 in comparison with Speedtest by Ookla For most countries, *adNPM* results closely match Ookla's data, except for Italy, Spain, Mexico, South Africa, Chile, and Guatemala where discrepancies arise. However, if we turn to Opensignal and consider the market share of the mobile networks in each country, it allows us to determine the average download speed values. Figure 5 compares *adNPM* with Opensignal data (except for the Dominican Republic and Ecuador, as

Opensignal does not report information for these countries), showing closer alignment. This suggests Ookla may overestimate speeds, especially for medium and high-speed networks, while *adNPM* aligns better with Opensignal's average download speeds.

Specifically for Italy, based on Opensignal's Q3 2023 Mobile Network Experience Report [87] and market share [86], the average download speed values for the country's largest operators (Fastweb, Iliad, TIM, Vodafone, and WindTre) is ≈ 31 Mbps. This value closely aligns with *adNPM*'s measurements, with an absolute error of 3 Mbps. For Guatemala, Chile, and South Africa, average speeds respectively to do with the mobile operators of these countries [51, 53, 107] and market share [19, 66, 80] are 22, 25.5, and 29 Mbps. *adNPM* results reveal only slight differences within (1 Mbps): 21.15, 25.89, and 28.85 Mbps. In Spain, adNPM download speed is lower than Ookla's, however it is consistent with Opensignal, with an absolute error of 3 Mbps.

For Mexico, we provide detailed information in Appendix A.2. There appears to be a discrepancy between the Speedtest Global Index from Speedtest by Ookla and the speeds obtained when analyzing individual operators in Mexico using Speedtest by Ookla report [11]. This discrepancy is unique to Mexico and not observed in any other country, suggesting a potential error in the Speedtest Global Index calculation.

### 5.2 OS-level Download Speed

We compare download speeds between iOS and Android OS across our dataset, broken down by country. Figure 6 shows that iOS generally provides higher download speeds than Android in most countries, with the largest difference in Guatemala (iOS: 42.52 Mbps, Android: 19.03 Mbps), and the smallest in the United Kingdom (iOS: 39.60 Mbps, Android: 38.42 Mbps). However, in South Africa, Android outperforms iOS with a download speed of 27.35 Mbps versus 21.29 Mbps for iOS, which is an exception to the overall trend.

This difference can be attributed to iOS's high-end devices and efficient resource management, which leads to tighter control over applications and bandwidth allocation.

---

[3]Speedtest by Ookla does not report on Puerto Rico's cellular network but shows it had the fastest Caribbean download speed at 46.84 Mbps in Q2 2022 [73]. *adNPM* measurements are from October to December 2023.

Instead, Android's broader device range, from cheap low-performance to high-end, introduces fragmentation, impacting network performance. Furthermore, Apple's software update policies ensure that a majority of iOS devices work with the latest versions, which can improve efficiency during data transmissions [100].

### 5.3 Download Speed demographic analyses

Exploring potential download speed differences based on demographic attributes is valuable for identifying any disparities in social development [13]. Note that, to the best of the authors' knowledge, our *adNPM* methodology is the only one enabling this type of demographic analyses.

We filter our dataset by `userGender` and `userAge` attributes collected from our DSP. For gender, we split users into male and female groups. For age, we categorize users into generations 'Gen Z', 1997-2012; 'Millennials', 1981-1996; 'Gen X', 1965-1980; 'Baby Boomers', 1946-1964. We exclude 'Gen Alpha' (born 2013-) and 'Gen Silent' (-1945) due to age restrictions and insufficient sample sizes, respectively.

Figure 7 compares the average download speeds for men and women at country-level, revealing no significant differences between genders.

Figure 8 shows the empirical distribution (pdf) of download speeds by age group. The aggregated results for our complete dataset, all countries, indicate minimal variation in connectivity speed across different age demographics. Note that country-specific results, with at least 300 samples per generation, are detailed in Appendix A.3.

The results are valuable for social scientists as they show that connection performance does not seem to contribute to generational or gender social divides. Upon examining download speeds across generations, from 'Gen Z' and 'Baby Boomers', we find consistent results. Age does not appear to play a significant role in determining download speed or connectivity experience.

### 5.4 Download Speed per Mobile-Operator

We compute average download speed offered by major ISPs in eight countries, each with at least 300 samples per operator to meet the criteria outlined in Subsection 4.1. As indicated in Section 3, the provider information is obtained through the `deviceCarrier` attribute collected from our DSP. This approach allows us to cross-examine the differences between main internet service providers within each country and understand how these variations may impact user experience.

Figure 9 plots the average download speed performance of key mobile operators in each targeted country. The range between the fastest and slowest ISPs shows notable variability, with differences of 28.31 Mbps in Spain, 6.29 Mbps in Colombia, 14.71 Mbps in Chile, 34.30 Mbps in South Africa,

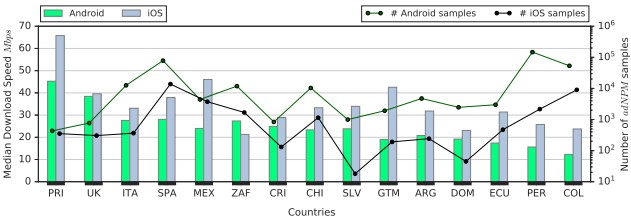

**Figure 6:** *left* y-axis: *adNPM* d/l speed by OS and country; *right* y-axis: number of ad campaign samples.

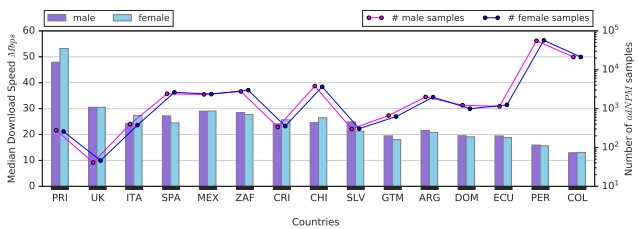

**Figure 7:** *left* y-axis: *adNPM* d/l speed by gender and country; *right* y-axis: number of ad campaign samples.

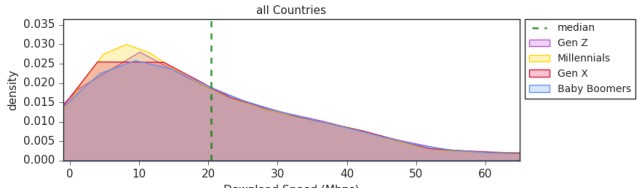

**Figure 8:** *adNPM* download speed distribution by generations (dashed line shows median $\mathcal{DS}$).

7.29 Mbps in Guatemala, 29.09 Mbps in Mexico, 15.59 Mbps in Peru, and 11.70 Mbps in Argentina.

This suggests that user experience can vary significantly based on the ISP, affecting access to services like HD streaming (see Figure 1), which require speeds over 15 Mbps. Users with the slowest ISPs in Peru, Mexico, or South Africa might struggle with these services, while Telcel, Movistar, Claro, and Vodacom report sufficient speeds. Instead, ISP choice has minimal impact in Chile, Colombia, and Guatemala due to narrow speed gaps. In Spain and Argentina, a single ISP leads with the fastest speeds. These findings align with Opensignal's results in its Mobile Network Experience Report [15, 50–53, 62, 82, 107].

## 6 RELATED WORK

Several efforts have leveraged the use of ads to perform large-scale network measurements. Our work shares Callejo et al.'s vision of using ads as a vehicle for passive data collection

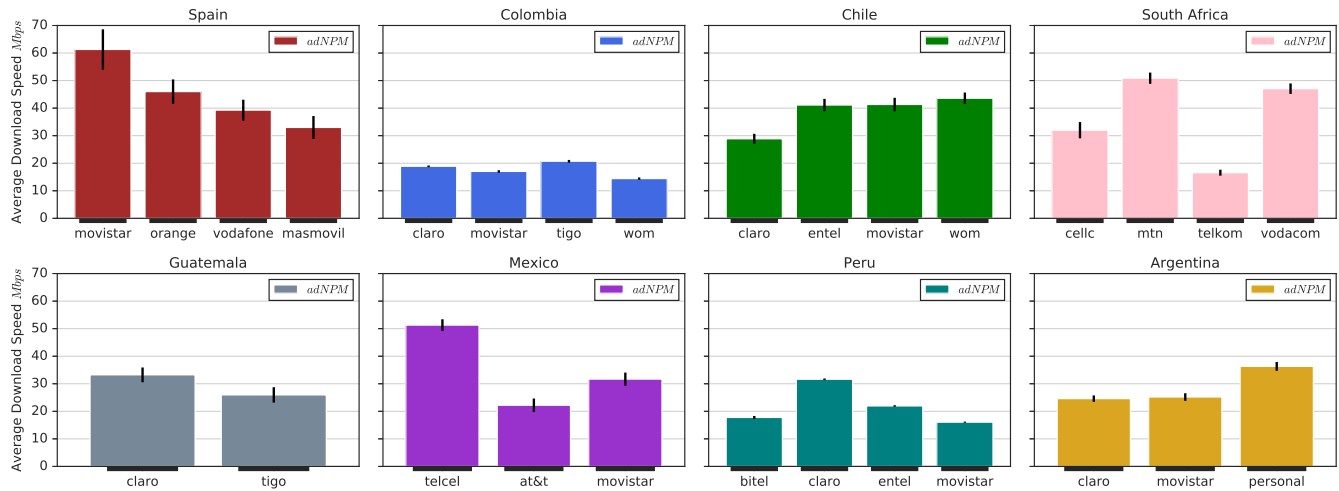

Figure 9: Performance of *adNPM* download speed across major Internet Service Providers (ISPs) by country.

[28]. However, Callejo et al. focus on measuring ISP coverage, ad execution time, and API support in browsers, whereas *adNPM* measures download speed bandwidth, which is significantly more challenging due to the need for large-scale, real-time accuracy in reflecting user experience across diverse network conditions. APNIC uses ads to estimate the quality of network connections, particularly through the measurement of DNS performance [58]. Similarly, in terms of injecting measurements into page loads, Odin, a CDN measurement system [26], uses active client-server measurements but differs from adNPM's ad-driven approach, which scales more cost-effectively across broader populations.

Most commercial speed tests, including those offered by ISPs [5, 32] and non-ISP entities [24, 48, 70, 95], use flood-based tools that saturate the bottleneck link through active measurements. These tools measure the maximum performance between a client and a test server using one or multiple concurrent TCP flows over a specific duration [49]. Although methodological differences in these tools affect speed measurements. NDT7 [69], a protocol from M-Lab using TCP BBR, tends to report more conservative speeds compared to Speedtest by Ookla [49, 71]. However, these tools often under-represent regions with access gaps, where users are less likely to conduct speed tests. *adNPM*, through passive data collection via online ads, addresses this by capturing performance variations across diverse geographic and demographic scenarios without requiring user participation.

Numerous studies have used data from tools like Ookla and M-Lab to evaluate network performance, such as Canadi et al.'s analysis of broadband in metropolitan areas [29] and the FCC's Measuring Broadband America project [35, 91]. Similarly, Goga et al. [55] and Sundaresan et al. [99] evaluated the accuracy of speed measurement tools in residential networks. While these studies provide valuable historical insights, *adNPM* addresses contemporary challenges by using the scalable and ubiquitous nature of digital advertising to measure download speed bandwidth.

*adNPM* offers a modern approach to achieve a goal set forth by measurement researchers almost two decades ago i.e., *deployment of measurement agents inside edge networks, generating regular test traffic of sufficient scale and diversity* outlined in Casado and Garfinkel's earlier work [30].

## 7 CONCLUSION

In this paper, we present *adNPM*, a novel technique for measuring download speeds by embedding measurement code within ads across web browsers and mobile apps. Unlike traditional tools, *adNPM* achieves large-scale, real-time, and unbiased measurements without any need for active user involvement. Through our experiments, conducted in both controlled lab environments and real-world scenarios across 15 countries, *adNPM* proves its ability to produce accurate results comparable to industry-standard tools like Speedtest by Ookla and Opensignal, but with significantly reduced data usage.

*adNPM*'s non-intrusive nature, deployed through ad campaigns, allows for the collection of large amounts of data from diverse demographics and geographies, offering detailed insights into the download speeds provided by major ISPs. Its ability to analyze download speeds based on demographic details and device OSs showcases its versatility and potential for targeted network performance assessments

*adNPM* emerges as a highly scalable, cost-efficient, and accurate tool for measuring mobile download speeds, paving the way for inclusive and realistic assessments of internet connectivity worldwide.

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

# A APPENDIX

## A.1 Ethical considerations

The research and experiments described in this paper have obtained the IRB approval of our institution through the Data Protection Officer, a member of the Ethics Committee responsible for approving projects and experiments with potential data protection implications.

Furthermore, to ensure responsible and non-intrusive data collection practices, ad campaigns were set up with a frequency cap of 1, meaning each user was only exposed to the experiment once. This prevented unnecessary or excessive data transfers that could disrupt the user's browsing experience. While our methodology involves a deliberate transfer of up to 65.72 MB of data to users' devices -6.35 MB at 5 Mbps; 15 MB at 15 Mbps; 24.25 M at 25 Mbps; 48 MB at 45 Mbps-this should be seen in the context of today's digital activities. The average user typically uses between 10 and 15 GB of data per month for various online activities [105], such as streaming, social network browsing, gaming, and other Internet activities. Therefore, the maximum consumption of 65.72 MB in our methodology represents a small fraction of the bandwidth consumed during routine online activities such as web browsing or email. Given the guarantee offered by the frequency cap of 1, the maximum bandwidth consumption of our experiments represents roughly just 0.45-0.65% of a user's monthly data usage.

Finally, to comply with privacy regulations and respect user preferences, we refrained from collecting data from users who enabled Do Not Track (DNT). By excluding such users from our sample, we respected their explicit choice to opt out of tracking and emphasized our commitment to ethical data collection practices.

## A.2 Mexico's Internet Download Speed

We provide supplementary assessments of internet download speed measurements in Mexico. We compare our results with reports from Speedtest by Ookla, OpenSignal, and nPerf. Additionally, we include comparisons with other countries and examine the representativeness of the samples, as well as possible biases in the data collected.

*A.2.1 Country-level Performance Comparison for Mexico.* Mexico's measurements of emphadNPM show discrepancies concerning Ookla's outcomes. Nevertheless, if we stick to Ookla's Mexico Market Report [11], they deliver an overview of download speed performance for the largest mobile operators. If we extrapolate the results to the market share of mobile operators in Mexico for 2023 [43], Speedtest by Ookla's median download speed for Mexico would be $\approx 34$ Mbps, which differs significantly from the aggregate values reported by Speedtest by Ookla in its Speedtest Global Index, 25.37 Mbps.

Furthermore, since Ookla also provides Ookla Market Reports for Guatemala [10], Peru [12], Argentina [6], Costa Rica [7], Ecuador [8] and El Salvador [9] at the operator level [1, 19–22, 108], we repeat the analyses and verify, on the basis, that the median download speed values shared by Speedtest by Ookla in its Speedtest Global Index, are aligned with the download speed values calculated according to the results of the download speeds of the country's large operators and the market share for the same period of the year. Respectively, they would be $\approx 30, 5$ Mbps, $\approx 18, 00$ Mbps, $\approx 24$ Mbps, $\approx 27$ Mbps, $\approx 22$ Mbps, and $\approx 27$ Mbps. These network performances in download speed would substantiate the results of our *adNPM* measurements in Mexico.

OpenSignal's report for Q3 2023 [52] only includes the average download speed of two of Mexico's largest infrastructure based internet service providers, Telcel and AT&T. Based on their market share, the average value provided by OpenSignal would be $\approx 28.5$ Mbps.

Meanwhile, nPerf shares insights into the performance of top mobile operators in certain countries by doing several million tests and billions of mobile network coverage measurements per year to measure the quality of Internet connection. nPerf also provides its results by the average performance of download speed observed [79]. Based on their 2023 report [81] and the 2023 market share of mobile operators in Mexico, the average download speed value would be $\approx 31$ Mbps. If we compute the *adNPM* average download speed performance for comparison with nPerf, the average download speed would be 29.71 Mbps.

By analyzing our datasets we discover the prevalence of iPhone devices, which make up 45.87% of our dataset, despite Apple's market share in Mexico being 22.5% [96]. It stands in contrast to device sales data for Q2 2023, which places the iPhone 11, a device that does not support 5G connectivity, as the third best-selling device in Mexico [68]. It may suggest a slight bias towards lower download speeds for iOS compared to those speeds obtained through web browsers.

These findings suggest that the discrepancies identified in our data may be attributable to differences in data collection methodology between Speedtest by Ookla and OpenSignal, as well as the representativeness of the market share of mobile networks in each country. Speedtest by Ookla may be capturing a data sample that does not fully reflect the reality of the connectivity experience in some countries, while OpenSignal may provide a more accurate and representative perspective due to its focus on continuous and large-scale measurements.

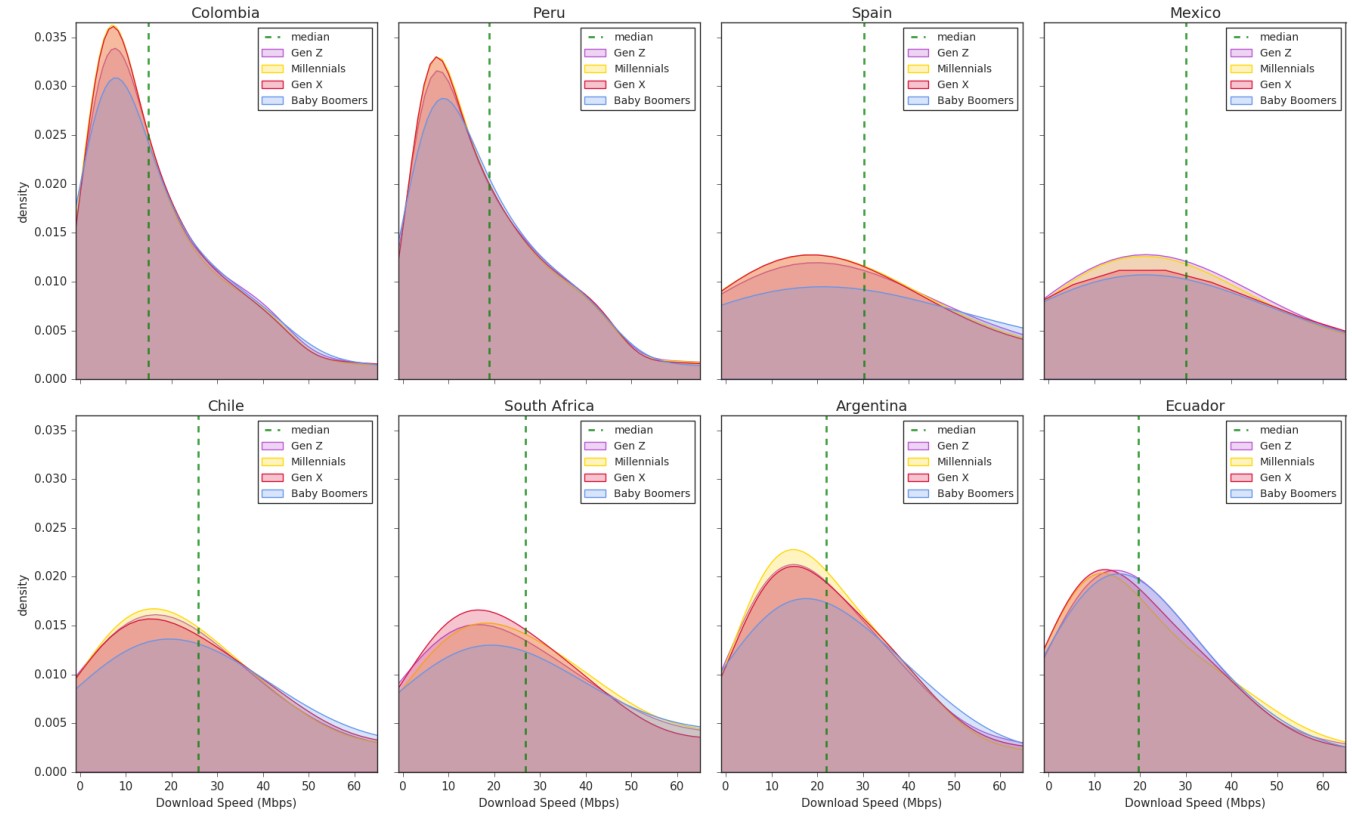

**Figure 10: *adNPM* download speed distribution by generations and countries (dashed line shows median $\mathcal{DS}$).**

## A.3 Download speed by generation and country

Figure 10 shows the the empirical distribution (pdf) for the download speed experienced by different demographic groups. We split the population into different age groups, as represented by the generations mentioned in Subsection 5.3. The figure shows the aggregated results for every country in which we have at least 300 samples for all considered generations. As detailed in Subsection 5.3, we find consistent results across all the evaluated countries. Both 'Gen Z' and 'Baby Boomers', and all generations in between, exhibited similar download speeds. Age does not appear to be a critical factor in determining the connectivity experience of download speed.

Received October 14th, 2024; revised December 1-14th, 2024; accepted January 20th, 2025

