# OpenReview forum: "Unveiling Network Performance in the Wild: An Ad-Driven Analysis of Mobile Download Speeds"
_ACM.org/TheWebConf/2025/Conference — WWW 2025 Poster_

### Official Review · Reviewer_p69R · 2024-11-26

**Novelty:** 4
**Technical Quality:** 5

**Review:**

#Paper summary

This paper targets the problem of measuring network performance. It proposes adNPM, which measures mobile network performance using embedded measurement code in advertisements displayed on web browsers and mobile apps. adNPM operates in an "active" manner, which requires no user participation. adNPM is evaluated through controlled lab tests. It is also applied to analyze real-world networks in 15 countries.

#Strengths

1. adNPM shows performance that aligns closely with established/commercial tools, but consumes significantly less data than they do.

2. adNPM well leverages existing ad infrastructure. This makes it easy to deploy at scale without requiring additional resource deployment.

3. Large-scale real-world analyses are conducted, demonstrating adNPM's effectiveness and revealing some insights about current mobile network measurement.

#Weaknesses

1. The contribution of this paper needs to be more clear. It seems to me the paper claims both adNPM approach and the landscape analysis as its contribution. I have some concerns regarding the former. Please see below for details.

2. How the required information, including location, demographics, type of device (mobile vs. fixed), operating system & browser, etc., is collected? In this process, how to enforce privacy protection and ethics?

#Detailed comments

My main concern is around adNPM as a measurement approach:

- Privacy and ethics: The use of embedded ads for data collection could raise privacy issues, particularly if user consent and data handling practices are not adequately addressed. The study of this paper has been well covered by IRB. However, when adNPM is deployed as a tool, its "active" mode (passive from end user perspective, and active from the ad owner) may raise privacy and ethics concerns.

- adNPM's reliance on ad campaigns may also limit its real-world applicability. Ad networks are usually restricted in regions or platforms, which may lead to unbalanced distribution of its deployment. In addition, adNPM's data collection is subject to restrictive ad-blocking practices. The paper states that "Mobile app measurements are not affected by Google’s policies [54]". This needs more detailed elaboration.

- Although adNPM's measurement consumes less data, it still introduces much more latency to the ad. This is another weakness that hinder adNPM from being a practical measurement tool.

Considering these noticeable weaknesses of adNPM as a measurement tool, I would recommend the paper focus its contribution on the real-world measurement.

**Questions:**

1. How the required information, including location, demographics, type of device (mobile vs. fixed), operating system & browser, etc., is collected? In this process, How does adNPM ensure user privacy and comply with data protection regulations like GDPR? Are data owners informed the data collection, and how is consent obtained from them?

2. If adNPM is to be used as a general tool, how does it address ethical concerns about embedding measurement tools in ads without active user participation?

3. Ads are typically personalized. Does this lead to any unbalance in the data collected? If yes, how does adNPM address potential unbalanced distribution?

4. How does adNPM address ad blockers? What impact would ad policies have on the effectiveness of adNPM?

**Reviewer Confidence:**

3: The reviewer is confident but not certain that the evaluation is correct

**Scope:**

3: The work is somewhat relevant to the Web and to the track, and is of narrow interest to a sub-community

---

### Official Review · Reviewer_nCZH · 2024-11-29

**Novelty:** 5
**Technical Quality:** 5

**Review:**

**Summary:** This paper introduces **adNPM**, a novel method for measuring mobile network download speeds by embedding lightweight measurement scripts in online advertisements. This approach enables large-scale, cost-effective data collection across diverse demographics without user participation. Validated through lab tests and real-world deployments, adNPM demonstrates accuracy comparable to traditional tools like Speedtest, while significantly reducing data usage.

### **Strengths**

- Practical Relevance
- Feasible Methods
- Detailed Studies

### **Weaknesses**

- Potential Ethical Concerns
- Missed Illustration for adNPM workflow

### Overall Comments for authors

This paper addresses an important issue in real-world network measurement — specifically, measuring download speeds — through an innovative approach of embedding measurement scripts within advertising platforms. This novel methodology, adNPM, is both clever and effective. Based on experimental results from both controlled lab environments and real-world deployments across countries, the paper demonstrates that adNPM offers high measurement efficiency. Compared to traditional tools, it achieves better results at a lower cost. The evaluation is comprehensive, considering a variety of influencing factors, and the analysis of measurement results is thorough. However, there are a few points where further improvements can be made, as outlined below.

First, while the authors have made significant efforts to address privacy concerns, there are still some aspects that could be clarified and enhanced. First, although the paper mentions excluding users who have enabled Do Not Track (DNT), it does not provide sufficient details on the anonymization and encryption methods used to protect user data during collection, storage, and transmission. Further clarification on how personal data is anonymized or aggregated would strengthen the privacy safeguards.
Moreover, the embedded code execution for measuring network performance might raise concerns regarding user consent and transparency. While the authors have stated that the research obtained IRB approval and ad campaigns were set up with a frequency cap, ensuring that users were not subjected to excessive data transfer, there is no clear indication that users were explicitly informed or provided consent to participate in the experiment. This is particularly relevant as the methodology involves executing code within ads, which could potentially be seen as an intrusive data collection practice if users are not fully aware of it. Although the authors excluded DNT users, the paper does not elaborate on whether all users were properly notified about the measurement and whether they had the opportunity to opt in or opt out.
Additionally, the paper does not discuss the data storage duration or data usage beyond the experiment, which is an important consideration for ensuring compliance with privacy regulations like GDPR. Lastly, if the data is being transmitted across borders, the authors should address any potential issues related to cross-border data transfer and ensure that the methodology complies with international data protection laws. These additional details would help ensure that the research adheres to the standards of privacy protection.

The second suggestion is that it would be beneficial to include an illustrative diagram of the workflow for adNPM. This would provide a clearer visual understanding of how the measurement process operates, making it easier for readers to follow the steps involved in the methodology.

### **Comments on Rigor**

The paper demonstrates a high level of rigor in its experimental design. It incorporates both controlled laboratory environments and in-the-wild experiments, providing a well-rounded assessment of the methodology's effectiveness in different settings. Additionally, the authors address potential issues that could affect measurement accuracy, such as ad blockers, browser settings, background traffic, and low bandwidth conditions, and provide an analysis of their impact on measurement reliability.

The measurement section also includes a thorough examination of how various factors—such as geographic location, user identification, operating system, and mobile operator—affect download speed results. This comprehensive analysis of different conditions and influencing factors ensures that the study accounts for a wide range of real-world scenarios, making the overall methodology robust and well-supported. The overall approach is therefore quite rigorous, taking into consideration both technical and environmental variables that could impact the measurement outcomes.

### **Comments on Novelty**

This paper introduces a novel approach by using ad-embedded scripts to measure mobile network download speeds. The idea of leveraging advertising platforms to unobtrusively collect network performance data is highly innovative, as it avoids the need for user participation or specialized infrastructure. This method allows for large-scale, cost-effective measurements across diverse user groups, with the added benefit of targeting specific demographics, operating systems, and geographic regions.

### **Comments on Presentation**

As I mentioned earlier, the paper is generally well-written. However, it would benefit from the inclusion of a workflow diagram for adNPM.

**Questions:**

### **Questions**

1. How personal data is anonymized or aggregated during collection, storage, and transmission?

2. Could the authors elaborate on whether all users were properly notified about the measurement and whether they had the opportunity to opt in or opt out?

3. How was the data handled or stored beyond experiment? How did the authors handle potential cross border data transmissions if any?

**Reviewer Confidence:**

3: The reviewer is confident but not certain that the evaluation is correct

**Scope:**

4: The work is relevant to the Web and to the track, and is of broad interest to the community

---

### Official Review · Reviewer_Uixt · 2024-12-01

**Novelty:** 1
**Technical Quality:** 1

**Review:**

This is totally outside my research area.

**Questions:**

-

**Reviewer Confidence:**

1: The reviewer's evaluation is an educated guess

**Scope:**

4: The work is relevant to the Web and to the track, and is of broad interest to the community

---

### Official Review · Reviewer_sV3Z · 2024-12-01

**Novelty:** 5
**Technical Quality:** 5

**Review:**

This paper describes adNPM, a new technique for measuring download speeds through embedded measurement code in advertisements.
The idea of using advertisements for network measurement appealed to me, and experiments have shown that this method of measurement can both achieve similar accuracy to other methods and provide fine-grained results that other methods cannot. The paper overall is clear and easy to understand.

**Questions:**

1. My main concerns about using ads for network measurement lie in the issue of the impact of measurement on the user experience. In my experience, the load time at the start of a video is critical to the user experience. Although the appendix mentions that only one measurement was taken per user, it still seems that if the measurements were routinized it would greatly affect the user experience. Can you provide quantitative impact of measurement on user experience?
2. Does the fact that adNPM relies on DSPs for delivering ads adversely affect the distribution of the measurement sample and the results?

**Reviewer Confidence:**

2: The reviewer is willing to defend the evaluation, but it is likely that the reviewer did not understand parts of the paper

**Scope:**

4: The work is relevant to the Web and to the track, and is of broad interest to the community

---

### Official Review · Reviewer_zshH · 2024-12-01

**Novelty:** 3
**Technical Quality:** 3

**Review:**

PROS:

 - I would like to thank the authors for the submission. This is an interesting paper that presents a system for measuring download speed by using adds loaded into the user's web browser or applications.
 - The paper is well-written and the methodology used is clear.

CONS:

 - The results are interesting at best, but vey limited due to the nature of the data available using the ad-driven approach.
 - The dataset lacks many features that could be insightful to explain the results, such as connection technologies (5G, 4G, etc) or signal strength.
 - The country-level analysis does not account for the granularity required to understand the specifics of different regions.
 - Some of the analysis (for example, effect of age or gender on download speeds) are unnecessary given it was very clear these factors would not have any effect at all (as concluded nonetheless after a lengthy discussion).
 - The statical analysis is vey limited (Spearman’s rank only, and very occasional correlations analyzed due to the nature of the data).
 - In general, despite the convenience provided by the ad-driven approach for download speed measurement, the nature of the data collected leads to a much more superficial and less insightful analysis compared to the one that can be performed using traditional measurements.

DETAILED COMMENTS:

Although the authors contrast their measurement techniques with more established speed tests, the paper lacks a more thorough discussion on any bias that might affect the results. For example, measurements using adds tend to be performed universally, regardless of user or his/her current activities (e.g., he or she might be driving) whereas speedtests run in particular occasions only, and are typically performed by more tech-savvy users interested in measuring their connection speed.

One of the limitations of this work is that it does not measure upload speeds. Although the authors minimize this drawback, upload speed is a very important metric for the most demanding applications mentioned in Fig. 1, such as online gaming.

Authors should provide a list of contributions in the introduction to set this work apart from the existing literature.

Section 3.3 presents the methodology validation, in which you say you emulate 22 connection speeds, ranging from 2 to 110 Mbps. How were connection speeds emulated? What was the emulation setup used?

In the experiment setup, you mention that the "display video ad is hosted on TAPTAP’s CDN (Content Delivery Network) to ensure proximity to the device and reduce potential artifacts from distant server downloads". How to ensure data measured with different CDN servers is comparable? What other biases the approach might be subject to?

You say you use Sonata’s targeting capabilities to extract demographic data (age, gender). Why would this be relevant? Differently from mobile vs. fixed or OS type, these factors should not affect download speed. As concluded after a lengthy discussion, they indeed do not affect performance and perhaps the space in the paper could have been used to present more insightful conclusions.

How does the lack of connection technology (5G, 4G, etc) or signal strength limit the study? Also, the analysis does not consider time of the day in the measurements, as it is known that during business hours the network might be more congested and affect performance. Also, the paper seems to consider country level information only, without taking into account measurements in more crowded or sparse regions.

Figure 4 that compare adNPM with Ookla shows very discrepant results without explanation. You say that "this suggests Ookla may overestimate speeds, especially for medium and high-speed networks", but you do not provide further explanation.

In Section 5 you use Spearman’s rank correlation coefficient for some of the analysis. Why was this correlation coefficient chosen instead of Pearson, Phi or other coefficients?

The country-level analysis might be subject to its own bias as well. Even within a country, speeds might vary drastically from region to region. This is not accounted for in this study due to the granularity of the data collected.

Section 5.2 concludes that iOS generally provides higher download speeds than Android. Why is that? What could justify this conclusion? Perhaps iOS devices tend to be newer models with advanced technologies (5G connection and more memory) but unfortunately the analysis does not allow further insights to confirm this hypothesis due to the limitations of the dataset.

The download speed per mobile operator provides a very superficial analysis. Could this be correlated with other variables, such as the device types or connection technology of the measurements? This is not investigated in the paper. What other correlations could explain these findings?

**Questions:**

- What types of bias might influence the direct comparison of the ad-driven approach presented in this paper and traditional speedtests? (The ad-driven approach can perform measurements universally, whereas speedtests tend to be performed in more particular occasions).

 - Despite the convenience of the ad-driven approach for measurement analysis, the nature of the dataset limits the types of analysis. How does the lack of connection technology (5G, 4G, etc) or signal strength limit the study?

 - It is known that during business hours the network might be more congested and affect performance. How is this accounted for in the study? Moreover, measurements taken in more crowded or sparse regions might show very different results that are not captured by a country-level analysis. How do these factors affect the conclusions of the paper?

 - What could explain the discrepancies between adNPM and Ookla in Figure 4?

 - Why was Spearman’s rank correlation coefficient chosen instead of Pearson, Phi or other coefficients?

 - In the intro, 1st paragraph, you say: "the architecture of Internet services has evolved towards a centralized infrastructure". I am not sure what you mean. The Internet is heavily decentralized and not centralized.

 - In the intro, 4th paragraph, you say that your solution is a "valuable tool for assessing mobile network performance". This is a bit confusing, since it seems that you also claim the approach can be used for non-mobile measurements. Why do you mention only mobile? Why not desktop?

 - It would be interesting to know how the authors intend to improve further this paper in terms of future work, which is not mentioned in the conclusion.

**Reviewer Confidence:**

4: The reviewer is certain that the evaluation is correct and very familiar with the relevant literature

**Scope:**

2: The connection to the Web is incidental, e.g., use of Web data or API